# Transient Global Amnesia (TGA): Is It Really Benign? A Pilot Study on Blood Biomarkers

**DOI:** 10.3390/ijms26062629

**Published:** 2025-03-14

**Authors:** Fabio Rossini, Tobias Moser, Michael Unterhofer, Michael Khalil, Rina Demjaha, Cansu Tafrali, Maria Martinez-Serrat, Jens Kuhle, David Leppert, Pascal Benkert, Johannes A. R. Pfaff, Eugen Trinka, Slaven Pikija

**Affiliations:** 1Department of Neurology, Neurocritical Care and Neurorehabilitation, Christian Doppler University Hospital, Paracelsus Medical University and Center for Cognitive Neuroscience, European Reference Network EpiCARE, 5020 Salzburg, Austria; 2Department of Neurology, Medical University of Graz, Auenbruggerplatz 22, 8036 Graz, Austria; 3Multiple Sclerosis Centre and Research Center for Clinical Neuroimmunology and Neuroscience (RC2NB), Department of Biomedicine, University Hospital and University of Basel, 4031 Basel, Switzerland; 4Multiple Sclerosis Centre and Research Center for Clinical Neuroimmunology and Neuroscience (RC2NB), Department of Clinical Research, University Hospital and University of Basel, 4031 Basel, Switzerland; 5Department of Neurology, University Hospital and University of Basel, 4031 Basel, Switzerland; 6Department of Neuroradiology, Christian Doppler University Hospital, Paracelsus Medical University of Salzburg, 5020 Salzburg, Austria; 7Neuroscience Institute, Christian Doppler University Hospital, Paracelsus Medical University and Center for Cognitive Neuroscience, 5020 Salzburg, Austria

**Keywords:** neurofilament light chain, glial fibrillary acidic protein, Simoa, central nervous system, memory disorders, prognosis, TGA

## Abstract

We aimed to determine whether transient global amnesia (TGA) is associated with alterations in central nervous system (CNS) injury biomarkers—serum neurofilament light chain (sNfL) and serum glial fibrillary acidic protein (sGFAP). In a prospective cohort of TGA patients, blood samples were obtained within 24–48 h of TGA onset (t0) and 6 weeks thereafter (t1). We assessed sNfL and sGFAP levels using the highly sensitive single-molecule array assay and calculated Z-scores adjusted for age, gender, and body mass index (BMI). Demographics, electroencephalography (EEG), and cerebral magnetic resonance imaging (cMRI) findings were also collected. A total of 20 patients were included (median age: 66 years, 70% women). No significant changes in sNfL or sGFAP levels associated with TGA at t0 and t1 were observed. Median sNfL Z-scores were 0.45 (interquartile range [IQR] −0.09, 1.19) at t0 and 0.60 (IQR −0.61, 1.19) at t1. Median sGFAP Z-scores were 0.27 (IQR −0.45, 0.76) at t0 and 0.44 (IQR −0.27, 0.75) at t1. Similarly, in the subgroup of patients with diffusion-weighted imaging (DWI)-positive hippocampal lesions (*n* = 5/20[25%]), no elevations in blood biomarkers were detected. Our pilot study on neurological blood biomarkers supports the benign nature of TGA, indicating that no CNS tissue damage occurs.

## 1. Introduction

Transient global amnesia (TGA) is a neurological syndrome characterized by the abrupt onset of anterograde amnesia, for episodic memory, resulting in the inability to form new memories. Patients typically exhibit repetitive questioning and may also experience varied degrees of difficulty recalling general or personal information (referred to as retrograde amnesia). During an episode of TGA, other cognitive functions, specifically semantic memory, remain fully intact [1,2]. TGA predominantly affects individuals between the ages of 50 and 80, with an estimated annual incidence rate of 32 cases per 100,000 individuals across this age group [3]. Although a single definite cause has not been determined, epidemiological and imaging data suggest several potential underlying pathophysiological processes, including migraine, vascular, epileptic, and psychogenic mechanisms [4]. The diagnostic approach focuses on excluding other conditions that may mimic TGA, such as epileptic seizures and strokes.

TGA is considered self-limiting and fully resolves, by definition, within 24 h [5]. In fact, several studies have demonstrated that patients with TGA do not have increased risks of mortality, epilepsy, cerebrovascular events, or dementia compared to age-matched controls [6,7,8,9,10,11,12]. However, TGA shares magnetic resonance imaging (MRI) features with acute cerebral ischemia [1,13] and has recurrence rates exceeding 20% [6].

While current diagnostic criteria rely on clinical presentation [5], recent advances in neuroscience have introduced neurological biomarkers that warrant a reassessment of TGA’s impact on the central nervous system (CNS). Neurofilament light chain (NfL) and glial fibrillary acidic protein (GFAP) are cytoskeletal proteins specific to the pathology of neurons and astrocytes, respectively, that correlate with brain tissue damage associated with various conditions, including inflammatory, degenerative, vascular, and traumatic brain injuries [14,15,16,17,18,19]. Among the quantitative measurement methods for soluble biomarkers, single-molecule array (Simoa^®^, Quanterix, Billerica, MA, USA) immunoassay shows an 126-fold and a 25-fold increased sensitivity compared to ELISA and electrochemiluminescence assays, respectively, thus enabling reliable and reproductible measurement of neuroaxonal injury via blood sampling [14,15]. In fact, serum NfL (sNfL) assessed by Simoa^®^ accurately reflects CSF levels in both individuals with neurological disorders and healthy controls, offering practical diagnostic approaches for neurological disorders [15].

Given its suggested benign nature, despite the most often found MRI lesions in CA1, we conducted an exploratory pilot study to test the hypothesis that TGA occurs without any measurable increase in the biomarkers sNfL and sGFAP.

## 2. Results

In this TGA cohort of 20 patients, the age ranged from the seventh to the eighth decade (median 66 years; range 61–75 years), with a clear female predominance (14/20; 70%, Table 1). Among cerebrovascular risk factors, arterial hypertension was the most frequent (11/20; 55%). The most frequent pathologies in the medical history were stroke and migraine. TGA recurred in 20% of the patients, and about 70% of the patients had a specific trigger. The median clinical duration was 6 h (range 3–9.5 h). MRI and EEG were performed one to two days after TGA onset. In total, 5 out of 20 patients (25%) exhibited a DWI-positive hippocampal hyperintensity (Table 2). The most common associated neuroradiological finding was leukoaraiosis. EEG was normal in 17 out of 20 patients, while three cases showed modest left-sided temporal theta slowing. None of the participants had undergone EEG prior to this study (Table 3).

Baseline sNfL values were 14 pg/mL (IQR 12–19), and baseline sGFAP values were 142 pg/mL (IQR 118–167). These levels did not significantly differ at the 6-week follow-up, with sNfL at 15 pg/mL (IQR 12–23) and sGFAP at 142 pg/mL (IQR 108–170, Table 2). Percentile and Z-score analyses also showed no significant differences (Table 2, Figure 1, Appendix A and Appendix A).

In the DWI-negative group, sNfL levels were 15 pg/mL (IQR 13–21) at baseline (t0) and increased to 23 pg/mL (IQR 19–25) at the 6-week follow-up (t1); however, these changes were not statistically significant (all *p*-values > 0.1; Table 4). In the DWI-positive group, sNfL levels remained stable, with 14 pg/mL (IQR 12–17) at both t0 and t1 (all *p*-values > 0.1, Figure 1A). sGFAP levels in the DWI-negative group were 128 pg/mL (IQR 123–133) at t0 and increased to 185 pg/mL (IQR 130–226) at t1. Conversely, in the DWI-positive group, sGFAP levels were 149 pg/mL (IQR 116–171) at t0 and declined slightly to 139 pg/mL (IQR 116–167) at t1. All changes were not statistically significant (Figure 1B).

Comparisons of sNfL and sGFAP levels between DWI-positive and DWI-negative groups revealed no significant differences. For sNfL, the *p*-values were as follows: DWI status (positive vs. negative), *p* = 0.821; timepoint of measurement (t1 vs. t0), *p* = 0.334; and the interaction between DWI status and timepoint of measurement, *p* = 0.429. Within-group analyses yielded *p*-values of 0.561 for the DWI-negative group and 1.000 for the DWI-positive group.

For sGFAP, the *p*-values were DWI status (positive vs. negative), *p* = 0.743; timepoint of measurement (t1 vs. t0), *p* = 0.821; and their interaction between DWI status and timepoint of measurement, *p* = 0.110. Within-group *p*-values were 0.561 for the DWI-negative group and *p* = 0.625 for the DWI-positive group. When comparing sNfL Z-score values between DWI-positive and DWI-negative groups, no significant differences were observed. The *p*-values were as follows: for DWI status (positive vs. negative), *p* = 0.684; for timepoint of measurement (t1 vs. t0), *p* = 0.245; and for the interaction between DWI status and timepoint of measurement, *p* = 0.318. Similarly, when comparing sGFAP Z-score values between DWI-positive and DWI-negative groups, no significant differences were found. The *p*-values were as follows: for DWI status (positive vs. negative), *p* = 0.684; for timepoint of measurement (week 6 vs. index measurement), *p* = 0.245; and for the interaction between DWI status and timepoint of measurement, *p* = 0.318.

Spearman correlation analysis revealed no association between the duration of TGA (in hours) and biomarker levels. There was no significant correlation between duration and sNfL Z-score (ρ = 0.06, *p* = 0.801), nor between duration and sGFAP Z-score (ρ = −0.01, *p* = 0.966; Appendix A).

## 3. Discussion

In this study, we challenged the assumption that TGA is a benign condition by assessing serum biomarkers of neuronal damage to provide a new perspective on the syndrome’s potential long-term implications. Our hypothesis was supported by the evidence that subtle, mostly subjective neuropsychological abnormalities may persist even days to months after the index TGA event [20]. Moreover, TGA is frequently associated with DWI-positive lesions on cMRI, which are the typical signs of acute cerebral ischemia [1,13]. We found no evidence that TGA causes neuroaxonal or astrocytic damage as assessed by sNfL and sGFAP. It is also noteworthy that no increase was observed in patients with DWI-positive lesions on cMRI performed within 48 h of the index event.

The diagnosis was based on Caplan and Hodges and Warlow criteria [5,21], and our patient group exhibited similar demographic features, comorbidities, and triggers, as well as neuroradiological, electroencephalographic, and neuropsychological findings typically observed in patients with TGA. Therefore, we believe that, despite the small sample size, our study population is representative.

While the pathophysiology of TGA remains unknown, its clinical presentation strongly suggests a bilateral dysfunction of the mesiotemporal lobes, including the hippocampi. This hypothesis aligns with findings from cMRI, which often show T2 hyperintensity and DWI-positive lesions, predominantly affecting the CA1 segment of the hippocampi. Since DWI alterations occur in the acute phase of cerebrovascular events, and cerebral ischemia has been proposed as a potential cause of TGA [20,22], we will use acute ischemic disorders as a model to interpret our findings.

Although sNfL levels peak days to weeks after a stroke and then remain elevated for 3–6 months depending on the specific study settings [23,24,25], a rise has been observed as early as the first day following an acute cerebrovascular event [26,27,28]. Therefore, with our approach of sampling 24–48 h after TGA onset, we would likely have detected increased sNfL levels if neuroaxonal damage had occurred. However, the Z-scores at t0, which adjust the raw values to a large healthy control population, were <1.5, indicating no increased values using this commonly used cutoff. We also found no changes in sNfL at t1, when sNfL would have already plateaued during ischemic events, indicating again that no cerebral damage had occurred. Therefore, given the kinetics of sNfL in stroke, we can argue that any potential ischemic mechanism underlying TGA is not sustained enough to cause axonal damage. Moreover, since sNfL assessed with Simoa^®^ is highly sensitive to other causes of brain tissue damage, including suggested mechanisms responsible for TGA like epileptic events [29,30,31], we can reasonably assume that, regardless of the underlying biology, TGA is not associated to neuroaxonal pathology. We observed minimal fluctuations in the blood biomarkers both between groups and intraindividually, which is consistent with the literature and supports our conclusion [32,33].

Compared to sNfL, the kinetics of the astrocytic biomarker sGFAP in cerebrovascular disorders are less well-defined. Evidence suggests that sGFAP levels rise in the hyperacute phase of cerebrovascular events, i.e., within 24 h of stroke onset [34,35]. Moreover, sGFAP appears to correlate with stroke severity and size [34]. A recent study reported elevated sGFAP levels within the first days after lacunar infarcts but, importantly, no elevation at 3 months post-stroke [35], suggesting that sGFAP increases during the acute phase and returns to baseline faster than sNfL. We found no increased sGFAP during the acute TGA phase. Noteworthy, the median sGFAP levels in our TGA cohort were lower compared to the patients with small vessel strokes [35]. Moreover, the Z-scores among our cohort were not increased and showed no dynamics. Therefore, our data regarding sGFAP also support the hypothesis of a benign nature of TGA.

Although TGA and seizures are clinically distinct entities, both involve CA1 hippocampal regions, allowing for a comparison of the extent of damage between these conditions. Several studies have explored the role of NfL in epilepsy, considering factors such as seizure duration, semiology, and etiology, which can influence biomarker levels [36]. Indeed, NfL and GFAP are significantly elevated in patients with epileptic seizures compared to those with psychogenic non-epileptic seizures and healthy controls [37]. Notably, sNfL increases more in status epilepticus [31,38] and sNfL and sGFAP in structural etiologies like stroke [29] or autoimmune encephalitis [39], suggesting that such conditions impact the extent of neuroaxonal damage differently. However, it can be challenging to determine whether the effects on biomarker kinetics are directly due to epilepsy or are consequences of the acute event leading to seizures. Also, there is conflicting evidence regarding whether the frequency of seizures increases neurological blood biomarkers [29,30]. Despite evidence of elevations of sNfL and sGFAP in epileptic patients, no study has specifically investigated the damage caused by a single, objectively documented seizure. In summary, the available data suggest that the increase in sNfL and sGFAP in seizures is greater than what we observed in TGA.

In addition to comparing TGA with conditions like seizures or strokes, it is also informative to consider its relationship with neurocognitive disorders that lack obvious structural lesions, such as primary psychiatric disorders (PPD). Indeed, sNfL has been useful in differentiating frontotemporal dementias from PPDs, with the latter showing a much less marked increase in sNfL [40,41]. PPD is an umbrella term encompassing various psychiatric disorders, including major depressive disorder (MDD), bipolar affective disorder (BPAD), and schizophrenia. The extent of NfL alteration in PPDs varies, typically showing a 1.2- to 2.5-fold increase compared to healthy controls, though less than in neurodegenerative diseases [42]. One study found no significant elevation in sNfL levels in individuals with their first episode of psychosis compared to healthy controls [43]. However, both sGFAP and sNfL were increased in chronic schizophrenia compared to first-episode psychosis [44]. The authors attributed these findings to longer disease duration and underlying degeneration in schizophrenia. In summary, PPDs are associated with an increase in sNfL and sGFAP, though acute, short-lasting episodes like the first episode of psychosis may exhibit kinetics similar to those of healthy controls.

To summarize, we did not detect any dynamics of neither sNfL (a biomarker for mid-term damage) nor sGFAP (indicative of hyperacute pathology) suggestive of brain damage occurring in TGA patients. This is ultimately supported by the Z-scores of both biomarkers in our cohort. Therefore, to the best of our knowledge, TGA can be considered a benign disorder of unknown etiology.

Our interpretations have some important limitations. A primary limitation is that we assessed sNfL and sGFAP levels shortly after the TGA episode and again 6 weeks later; thus, we are unable to characterize the trajectory of these biomarkers during the intervening period. Further studies are needed to enhance the understanding of the time course of serum biomarker levels. Additionally, the small brain region affected in TGA, in contrast to larger acute cerebrovascular events, may limit the ability to detect subtle increases in sNfL or sGFAP. Indeed, evidence shows that stroke volume correlates with both biomarker levels [25,34,35,45], which is an inherent limitation of this methodology. The binary nature of the MRI data collection (DWI-positive vs. DWI-negative) limited our ability to perform further correlation analyses in this pilot study. The hypothesis of the cerebral ischemia is one of the many which has been proposed to explain TGA, and further research is needed to explore alternative causes [20,22]. Finally, the findings from this pilot study with small numbers warrant validation in larger cohorts to confirm their generalizability and robustness.

## 4. Materials and Methods

For this prospective pilot study, 20 consecutive patients with TGA were recruited from the University Hospital of Salzburg, Austria, in 2024. The clinical diagnosis of TGA was made according to the criteria of Caplan and Hodges and Warlow [5,21]. Demographic data and clinical history were extracted from hospital records. Participants were excluded if any of the following conditions were present:Inability to undergo cerebral MRI (cMRI) or electroencephalography (EEG)Presence of acute intracranial damage, such as brain hemorrhage, traumatic brain injury, ischemic stroke, or transient ischemic attack within the preceding 3 monthsProlonged cardiopulmonary resuscitation lasting longer than 2 min within the past 12 monthsNeurosurgical operation within the last 6 monthsProgressive neurological disorders, such as neoplasm or dementiaMajor surgery, biopsy of a parenchymal organ, or significant trauma within the past 2 monthsLack of willingness or ability to participate in the follow-up examinations.

At the follow-up visit 6 weeks after the initial episode, we collected data concerning past medical history and potentially confounding variables, including COVID-19 infection, recent immunization, trauma, stroke history, alcohol consumption, nicotine use, diet, and prior occurrences of stroke and TGA.

### 4.1. sNfL and sGFAP

Serum samples were collected within 24–48 h of TGA onset (t0) and again 6 weeks later (t1) and centrifuged for 10 min with 3000× *g* at room temperature, and serum aliquots were stored at −80 °C until analysis. Samples were collected between 7 and 9 am for all patients to minimize potential diurnal variations. The concentrations of sNfL and sGFAP were measured using the Simoa^®^ Neurology 2-Plex B Kit on a Simoa^®^ HD-X analyzer (Quanterix, Billerica, MA, USA). This ultrasensitive immunoassay kit is designed for the quantification of NfL and GFAP in human serum. Analyses were conducted at the University of Graz (Austria) in accordance with the manufacturer’s guidelines. All samples were measured in duplicate using the same kit lot to ensure consistency. Measurements were performed on the same machine by laboratory personnel blinded to the sample identities and clinical data. The intra-assay coefficient of variation (CV) for all samples, including the quality control samples, was maintained below 20%.

Age-, gender- (for GFAP), and body mass index (BMI)-adjusted Z-scores for sNfL and sGFAP were calculated using large reference datasets [46,47]. Z-scores indicate the deviation from the healthy controls’ population mean; for example, a Z-score of 1 represents a concentration one standard deviation above the reference mean.

### 4.2. cMRI and EEG Assessment

Each patient underwent EEG and cMRI according to protocol within 24 to 72 h of TGA onset. EEG recordings were obtained using 19 sintered Ag/AgCl electrodes positioned according to the international 10–20 system on a Nihon Kohden EEG cap (model H564A). An additional electrocardiogram (ECG) channel was incorporated into the setup. When performed, hyperventilation and photostimulation were conducted following standard protocols. EEG data acquisition and analysis were carried out using Natus^®^ NeuroWorks^®^ 10 EEG Software (Middleton, WI, USA). Imaging was conducted using a 3 Tesla Achieva dStream system (Philips Medical Systems, Best, The Netherlands) at the Department of Neuroradiology, Christian Doppler University Hospital, affiliated with Paracelsus Medical University in Salzburg, Austria. A standardized protocol for identifying potential ischemic strokes was employed, incorporating diffusion-weighted imaging (DWI), fluid-attenuated inversion recovery (FLAIR), susceptibility-weighted imaging (SWI), time-of-flight (TOF), and T2-weighted sequences. Specifically, for the DWI sequence, spin- echo echo-planar imaging (EPI) diffusion imaging was utilized. This captured 28 slices with an echo time (TE) of 47 ms, a repetition time (TR) of 3051 ms, a field of view (FOV) of 230 × 230 mm, and voxel dimensions of 2.05 × 2.56 mm. The slice thickness was set at 4 mm with a 1 mm gap between slices. Diffusion-weighted images were acquired using four b-values (0, 333, 666, and 1000 s/mm^2^) with diffusion gradients applied along three directions.

### 4.3. Statistics

Baseline demographic, clinical, and biomarker data were summarized as medians with interquartile ranges (IQR) for continuous variables, reflecting their non-normal distributions, and as frequencies with percentages for categorical variables. Group differences were evaluated using appropriate statistical tests. Paired differences in sNfL and sGFAP levels between baseline (t0) and the 6-week follow-up (t1) within each group were analyzed using the Wilcoxon signed-rank test. Differences in biomarker levels between DWI-positive and DWI-negative groups were assessed using the Mann–Whitney U test for unpaired comparisons. To account for repeated measurements (t0 and t1) and group differences, generalized linear mixed models (GLMMs) were employed. Log-transformed sNfL and sGFAP values, adjusted for non-normal distributions, served as dependent variables. Fixed effects included timepoint of measurement (t0 vs. t1), DWI status (positive vs. negative), and their interaction to examine time-dependent group differences. Participant ID was included as a random effect to model within-subject variability, with a Gaussian distribution assumed for the transformed values. Additionally, correlation analyses were performed to assess the relationship between the duration of TGA (in hours) and biomarker levels. Given the non-normal distribution of duration, Spearman’s rank correlation coefficient was used to evaluate the association between duration in hours and sNfL Z-score, as well as duration and sGFAP Z-score. Statistical significance was defined as *p* < 0.05 for all analyses, with Bonferroni corrections applied to control for multiple comparisons. All statistical analyses were conducted using R (version 4.2.0), employing the lme4 package for GLMMs and ggplot2 for data visualization. Tables and figures were generated using the gtsummary and patchwork packages.

## 5. Conclusions

Our study on the neurological blood biomarkers sNfL and sGFAP in TGA patients supports the prevailing assumption that TGA is, in fact, a benign disorder not associated with brain tissue damage.

## Figures and Tables

**Figure 1 ijms-26-02629-f001:**
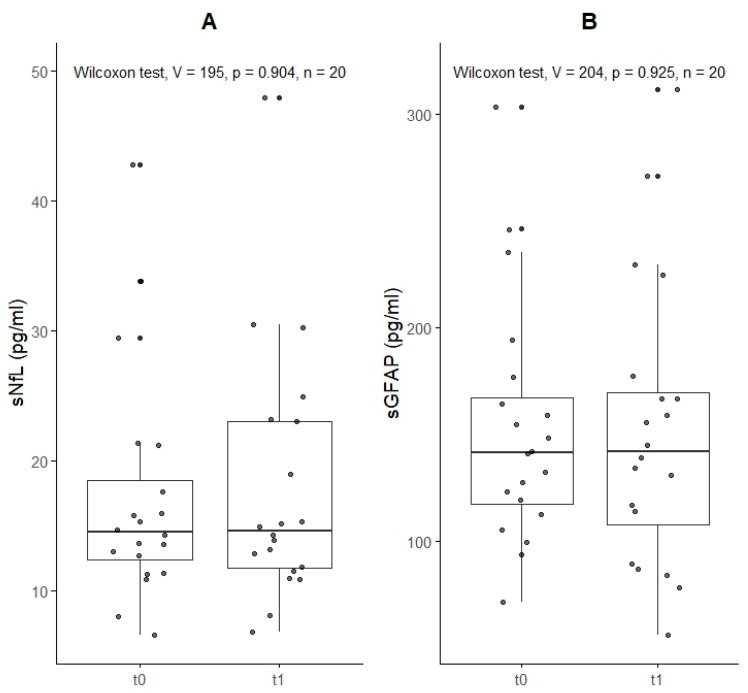
Serum neurofilament (sNfL, **Panel A**) and serum glial fibrillary acidic protein (sGFAP, **Panel B**) levels in 20 patients with emergent TGA, measured within 24 to 48 h of symptom onset (t0) and at 6 weeks (t1).

**Table 1 ijms-26-02629-t001:** Demographics, clinical, laboratory, and radiological characteristics of 20 patients with emergent transient global amnesia.

Demographic Features	*n* = 20 ^1^
Age at diagnosis of TGA	66 (61, 75)
Sex (Woman)	14 (70%)
TGA duration (h)	6.0 (3.0, 9.5)
BMI	25.9 (23.5, 27.4)
Comorbidities	
Arterial hypertension	11 (55%)
Hypercholesteremia	3 (15%)
Ischemic stroke in history	2 (10%)
Years since first stroke	5 (4, 6)
Atrial fibrillation	1 (5%)
Diabetes	1 (5%)
Smoking	2 (11%)
Medication	
Antithrombotic treatment (ASA all)	4 (20%)
TGA in the past medical history	4 (20%)
Years since first TGA	11 (7, 13)
Neurological diseases in past medical history (other than TGA)	
Migraine	2 (10%)
Vestibular neuritis	1 (5%)
Other	1 (5%)
None	16 (80%)
Vital signs at admission	
Systolic pressure (mmHg)	170 (155, 192)
Diastolic pressure (mmHg)	91 (88, 100)
Ear temperature (°C)	36.1 (36.0, 36.6)
Heart frequency (bpm)	81 (69, 90)
Laboratory values	
ESR (mm/h)	6.0 (4.0, 7.0)
CRP (mg/dL)	0.2 (0.1, 0.2)
HDL-C (mg/dL)	60 (50, 69)
LDL-C (mg/dL)	133 (107, 177)
hs-Troponin T (pg/mL)	8.5 (7.0, 11.8)
NT-proBNP (pg/mL)	95 (61, 169)
HbA1C (%)	5.4 (5.3, 5.4)
Triggers	
Emotional or stressful event	8 (42%)
Exercise of physical exertion	4 (21%)
Data not available	4 (21%)
Shower or bathtub	1 (5%)
None	2 (11%)
EEG findings	
Unilateral temporal slowing	3 (15%)
Normal	17 (85%)
EEG—days after TGA onset	1 (1, 2)
MRI—days after TGA onset	1 (0, 1)
MRI findings	
DWI hippocampal lesion	5 (25%)
DWI locations	
Left	1 (20%)
Left (CA1-Region), 3 mm	1 (20%)
Left temporal rostral (subiculum), 2 mm	1 (20%)
Right	1 (20%)
Right posterior	1 (20%)
Additional MRI findings	
Leukoaraiosis first grade	4 (21%)
Leukoaraiosis second grade	3 (16%)
Leukoaraiosis third grade	1 (5%)
Incidental corpus callosum hyperintensity	1 (5%)
DVA frontal right	1 (5%)
Mild atrophy	1 (5%)
MTA score 3	1 (5%)
Other	1 (5%)
None	5 (26%)

^1^ Median (IQR); n (%). Abbreviations (alphabetical): °C: degrees Celsius; ASA: Acetylsalicylic Acid; BMI: Body Mass Index; CA1: Cornu Ammonis 1; CRP: C-reactive Protein; DVA: Developmental Venous Anomaly; DWI: Diffusion-Weighted Imaging; EEG: Electroencephalography; ESR: Erythrocyte Sedimentation Rate; HbA1C: Hemoglobin A1c; HDL-C: High-Density Lipoprotein Cholesterol; LDL-C: Low-Density Lipoprotein Cholesterol; hs-Troponin T: High-Sensitivity Troponin T; MRI: Magnetic Resonance Imaging; MTA: Medial Temporal Lobe Atrophy; NT-proBNP: N-terminal Pro-B-Type Natriuretic Peptide; TGA: Transient Global Amnesia.

**Table 2 ijms-26-02629-t002:** Serum neurofilament (sNfL) and serum glial fibrillary acidic protein (sGFAP) levels measured in 20 patients with emergent transient global amnesia within 24 to 48 h of symptom onset (baseline, t0) and after 6 weeks (6 weeks, t1).

Summary of Biomarkers at Baseline (t0) and 6 Weeks (t1)						
Type	sNfL			sGFAP		
	Baseline, *n* = 20 ^1^	6 Weeks, *n* = 20 ^1^	*p*-Value ^2^	Baseline, *n* = 20 ^1^	6 Weeks, *n* = 20 ^1^	*p*-Value ^2^
Raw values, median (IQR) (pg/mL)	14 (12, 19)	15 (12, 23)	>0.9	142 (118, 167)	142 (108, 170)	>0.9
Percentile	68 (46, 88)	73 (27, 88)	0.8	61 (33, 78)	67 (40, 77)	0.8
Z-score	0.45 (−0.09, 1.19)	0.60 (−0.61, 1.19)	0.8	0.27 (−0.45, 0.76)	0.44 (−0.27, 0.75)	0.8

^1^ Median (IQR). ^2^ Wilcoxon rank sum exact test.

**Table 3 ijms-26-02629-t003:** Comparison of demographic, clinical, imaging (cMRI), electrophysiological (EEG), and laboratory data among 20 patients presenting with emergent transient global amnesia (TGA), stratified by the presence (5 cases) or absence (15 cases) of diffusion-weighted imaging (DWI) hippocampal positivity.

Characteristic	DWI—Positive, *n* = 5 ^1^	DWI—Negative, *n* = 15 ^1^	*p*-Value ^2^
Age at TGA diagnosis	66 (63, 74)	66 (61, 75)	>0.9
TGA duration (h)	6.0 (4.0, 7.0)	5.5 (3.0, 10.0)	0.9
Gender (Man)	0 (0%)	6 (40%)	0.3
BMI	28.7 (27.0, 30.5)	25.4 (23.0, 26.5)	0.12
Comorbidities			
Smoking	0 (0%)	2 (14%)	>0.9
Atrial Fibrillation	0 (0%)	1 (7%)	>0.9
Arterial Hypertension	3 (60%)	8 (53%)	>0.9
Diabetes	0 (0%)	1 (7%)	>0.9
LDL-C Hypercholesteremia	2 (40%)	1 (7%)	0.14
Ischemic stroke in history	1 (20%)	1 (7%)	0.4
Years since first stroke	7 (7, 7)	3 (3, 3)	>0.9
Neurological diseases in history (other than TGA)			0.10
None	3 (60%)	13 (87%)	
Migraine	0 (0%)	2 (13%)	
Other	1 (20%)	0 (0%)	
Vestibular neuritis	1 (20%)	0 (0%)	
Medication			
Antihrombotic therapy (all ASA)	3 (60%)	1 (7%)	0.032
TGA in history	1 (20%)	3 (20%)	>0.9
Years since first TGA	11 (11, 11)	8 (6, 11)	>0.9
Vital Signs at admission			
Systolic pressure at admission (mmHg)	172 (148, 201)	170 (158, 187)	>0.9
Diastolic pressure at admission (mmHg)	93 (89, 99)	91 (88, 99)	>0.9
Ear temperature at admission (°C)	36.05 (35.93, 36.23)	36.20 (36, 36.65)	0.4
Heart frequency (per minute)	83 (80, 85)	80 (68, 91)	>0.9
Laboratory Values			
BSG	6 (6, 6)	5 (3.50, 6.50)	>0.9
CRP	0.10 (0.10, 0.10)	0.20 (0.10, 0.20)	0.083
HDL-C	56 (51, 62)	64 (50, 73)	0.7
LDL-C (mg/dL)	142 (129, 204)	124 (101, 173)	0.5
hs-Troponin T	10.0 (6.0, 16.5)	8.0 (7.0, 11.5)	0.9
pro—BNP	169 (132, 207)	79 (41, 126)	0.3
HbA1C	5.40 (5.40, 5.40)	5.30 (5.25, 5.90)	0.8
Triggers			>0.9
Emotional or stressful event	2 (50%)	6 (40%)	
Exercise of physical exertion	1 (25%)	3 (20%)	
Shower or bathtube	0 (0%)	1 (7%)	
Data not available	1 (25%)	3 (20%)	
None	0 (0%)	2 (13%)	
EEG findings			>0.9
none	45(100%)	12 (80%)	
temporal slowing	0	3 (20%)	
EEG—days after TGA	2 (1, 2)	1 (0, 1)	0.094
Additional MRI findings			>0.9
none	3 (60%)	4 (27%)	
leukoaraiosis first grade	1 (20%)	3 (20%)	
leukoaraiosis second grade	1 (20%)	2 (13%)	
leukoaraiosis third grade	0 (0%)	1 (7%)	
Incidental corpus calosum hyperintensity	0 (0%)	1 (7%)	
DVA frontal right	0 (0%)	1 (7%)	
mild atrophy	0 (0%)	1 (7%)	
MTA score 3	0 (0%)	1 (7%)	
other	0 (0%)	1 (7%)	
MRI—days after TGA	1 (1, 2)	1 (0, 1)	0.4

Abbreviations (alphabetical): °C: degree Celsius; ASA: Acetylsalicylic Acid; BMI: Body Mass Index; CRP: C-reactive Protein; DVA: Developmental Venous Anomaly; DWI: Diffusion-Weighted Imaging; EEG: Electroencephalography; HbA1C: Hemoglobin A1c; LDL-C: Low-Density Lipoprotein Cholesterol; hs-Troponin T: High-Sensitivity Troponin T; MRI: Magnetic Resonance Imaging; MTA: Medial Temporal Lobe Atrophy; TGA: Transient Global Amnesia. ^1^ Median (IQR) ^2^ Wilcoxon rank sum exact test.

**Table 4 ijms-26-02629-t004:** Summary of biomarkers at baseline and 6 weeks. Serum neurofilament (sNfL) and serum glial fibrillary acidic protein (sGFAP) levels measured in 20 patients with emergent transient global amnesia and grouped by presence of hippocampal DWI—hyperintensity (DWI-positive, DWI-negative) within 24 to 48 h of symptom onset (baseline, t0) and after 6 weeks (6 weeks, t1).

Characteristic	sGFAP DWI—Negative Baseline	sGFAP DWI—Negative 6 Weeks	sNfL DWI—Negative Baseline	sNfL DWI—Negative 6 Weeks	sGFAP DWI—Positive Baseline	sGFAP DWI—Positive 6 Weeks	sNfL DWI—Positive Baseline	sNfL DWI—Positive 6 Weeks
*n* =	15	15	15	15	5	5	5	5
Raw value, median (IQR) (pg/mL)	149 (116, 171)	139 (116, 167)	14 (12, 17)	14 (12, 17)	128 (123, 133)	145 (87, 225)	15 (13, 21)	23 (15, 23)
Percentile	61 (33, 79)	58 (34, 77)	68 (45, 88)	70 (31, 88)	56 (53, 65)	71 (70, 76)	65 (50, 95)	83 (28, 89)
Z-score	0.28 (−0.45, 0.79)	0.20 (−0.43, 0.73)	0.47 (−0.14, 1.15)	0.52 (−0.52, 1.15)	0.15 (0.08, 0.39)	0.55 (0.52, 0.71)	0.39 (0.00, 1.60)	0.95 (−0.58, 1.23)

## Data Availability

The data that support the findings of this study are available on reasonable request by qualified researchers from the corresponding author (TM).

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
