# Peer review of "Transient Global Amnesia (TGA): Is It Really Benign? A Pilot Study on Blood Biomarkers"

_ijms, 2025, doi:10.3390/ijms26062629_

Round 1
Reviewer 1 Report
Comments and Suggestions for Authors
In this manuscript, the authors report that serum neurofilaments and GFAP do not significantly increase compared to the normal population and after several weeks following TGA, implying that TGA is indeed a benign condition, not accompanied by brain tissue damage. The study is clinically important, and the methodology is sound. It also adds to the knowledge on the clinical value of these newly introduced serum measurements of brain damage, which will probably become more widely used.
I do have several comments, regarding the interpretation of the results:
- The results should be compared not only to findings after strokes, but also to existing reports in the literature of Nfl and GFAP after seizures. Additionally, they can be compared with more "functional" events such as psychiatric attacks, like psychosis.
- Since some, albeit not statistically significant, differences (both decreases and increases) were found between T0 and T1 in the different groups and also for individual patients, the issue of fluctuations in these measurements in healthy individuals over time should be discussed in view of the available data from the literature and applied on the interpretation of the results
- Were the samples taken at specific time points during the day? GFAP has been reported to be higher before noon than in the afternoon
Reviewer 2 Report
Comments and Suggestions for Authors
This is an important manuscript concerning transient global amnesia (TGA), MRI diffusion, EEG, and blood biomarkers. Here are some suggestions to help improve the manuscript.
1) In order to evaluate the goodness of the biomarkers in characterizing TGA, the authors should do correlational analysis between TGA clinical scores with the biomarker scores and with the MRI scores. This will help define the relationship between the biomarkers and TGA. Show scatter plots of TGA clinical scores versus the biomarker scores and the MRI scores complete with regression line and Pearson’s correlation coefficient.
2) instead of breaking up the subjects into DWI-positive and DWI-negative groups, the authors should correlate DWI scores (from all of the subjects) with all of the characteristics shown in Table 3 (which is a great list of characteristics). In this way, there is no need for some artificial threshold in breaking up into different groups. By doing this correlation analysis, the authors can define the relationship between DWI and these important characteristics. I would raise my overall merit score to high if the manuscript could be revised.
Round 2
Reviewer 2 Report
Comments and Suggestions for Authors
The authors have addressed my previous concerns and the manuscript is now ready for publication.